# Econophysics and the Entropic Foundations of Economics

**DOI:** 10.3390/e23101286

**Published:** 2021-09-30

**Authors:** J. Barkley Rosser

**Affiliations:** Department of Economics, James Madison University, Harrisonburg, VA 22807, USA; rosserjb@jmu.edu

**Keywords:** econophysics, entropy, complex systems, ecological economics, urban–regional economics, income distribution, financial market dynamics

## Abstract

This paper examines relations between econophysics and the law of entropy as foundations of economic phenomena. Ontological entropy, where actual thermodynamic processes are involved in the flow of energy from the Sun through the biosphere and economy, is distinguished from metaphorical entropy, where similar mathematics used for modeling entropy is employed to model economic phenomena. Areas considered include general equilibrium theory, growth theory, business cycles, ecological economics, urban–regional economics, income and wealth distribution, and financial market dynamics. The power-law distributions studied by econophysicists can reflect anti-entropic forces is emphasized to show how entropic and anti-entropic forces can interact to drive economic dynamics, such as in the interaction between business cycles, financial markets, and income distributions.

## 1. Where Econophysics Came From

It has long been argued as for example by Mirowski [1] that economic theorists have drawn on ideas from physics, with an especially dramatic and influential example being Paul Samuelson’s Foundations of Economic Analysis [2] from 1947. However, while the influence of physics concepts in Samuelson, as well as many economists much earlier, was enormous and openly acknowledged, it was only much later that the term econophysics would be coined, reportedly at a conference in 1995 Kolkata, India [3] by H. Eugene Stanley, who as a longtime editor of Physica A has played a crucial role in publishing many papers that have been identified as representing and advancing this approach, with the term first appearing in print in 1996 [4]. Curiously when it came to define this multidisciplinary neologism, the emphasis given by Mantegna and Stanley [5] was not upon the ideas or specific theoretical methods involved, but rather on the people doing it: “the activities of physicists who are working on economics problems to test a variety of new conceptual approaches deriving from the physical sciences”.

This freshly defined approach involving physicists in particular, sometimes in conjunction with economists, quickly became a self-conscious cottage industry, even though arguably similar efforts had been going on for a long time, if not specifically by self-identified physicists, although some econophysicists have argued that an early inspiration for their work was Ettore Majorana in 1942 [6], whose untimely death gave him dramatic attention as he argued for the profound identity of statistical methods used in social sciences and physics. Important influences on the self-identified econophysicists included statistical mechanics [7,8] and also self-organized criticality models derived from models of avalanches [9] and earthquakes [10]. These approaches led to studies of many subjects in the early days, generally finding distributions that did not follow Gaussian patterns characterizable solely by mean and variance. These subjects included financial market returns [11,12,13,14,15,16,17,18], economic shocks and growth rate variations [19,20], city size distributions [21,22], firms size and growth rate patterns [4,23,24], scientific discovery patterns [25,26], and the distribution of income and wealth [27,28,29].

While the emerging econophysicists identified themselves as being physicists, an important impetus to their activities came from the intense discussions between economists and physicists at the Santa Fe Institute starting in the late 1980s [30,31]. While some of the economists defended existing economic theory, these discussions often emphasized dissatisfaction with its ability to explain empirical phenomena exhibiting non-Gaussian distributions with skewness and “fat tails” leptokurtosis [32,33,34]. While indeed most of the economists in these discussions disavowed some of the models developed by the econophysicists, the irony is that some of these models introduced by the physicists that could generate such higher moments as well as scaling properties were originally developed by economists, with the most important example of this being the Pareto distribution [35].

## 2. The Important Role of the Pareto Distribution

This important distribution that shows so many characteristics interesting to econophysicists was initially developed by the socio-economist Vilfredo Pareto in 1897 [35]. If *N* is the number of observations of a variable exceeding *x*, and *A* and *α* are positive constants, then
*N* = *Ax*^−*α*^.(1)

Scaling can be seen as:ln(*N*) = ln*A* − *α*ln(*x*),(2)
with it possible to stochastically generalize this by replacing *N* with the probability an observation exceeds *x*. The log–log form of this is conveniently linear.

Much like the more recent econophysicists, Pareto’s original focus was on income distribution, and he believed (inaccurately) that he had found the universally true value of 1.5 for *a*. In 1931, Gibrat [36] countered Pareto’s argument with the idea that instead income distribution followed the lognormal form of the Gaussian distribution that can arise from a random walk, first studied by Bachelier in 1900 [37], with Einstein adopting it to model Brownian motion [38]. However, further studies suggest that combining these two provides a better description of income distribution, with the upper end of the distribution showing a Pareto pattern and lower portions showing lognormal Gaussian forms [39,40,41,42].

As it was, the Gaussian random walk would come to dominate a great deal of the modeling of price dynamics and financial market dynamics, including the widely used Black–Scholes formula [43]. Ironically, this triumph of what became the standard economic approach was engineered by the physicist M.F.M. Osborne in 1959 [44]. His model of dynamic prices, with *p* as the price, *R* the price increase return, *B* as the debt, and *σ* as the Gaussian standard deviation, is given by:d*p* = *Rp*dt + *σp*d*B*.(3)

Nevertheless, parallel developments inspired by Pareto went on through the twentieth century, with some using the stable Lévy distribution developed in 1925 [45] as a generalization of Pareto’s distribution. Applications included looking at scientific discovery patterns [46] and city sizes [47]. A singular figure later in the century would be the father of fractal geometry, Benoit Mandelbrot [48,49], who directly posed the rival Pareto distribution as being able to model price dynamics [50] in 1963, in contrast to Osborne’s argument. In 1977, Iriji and Simon [51] applied this to firm size distributions, a finding generally ignored until verified by Rob Axtell in 2001 [52].

## 3. The Influence of Statistical Mechanics

Arguably, the earliest influence of physics on economics was due to Canard in 1801 [53], who posed supply and demand as being “forces” opposing each other in a physics sense. However, a more specific influence on conventional economics would be statistical mechanics, developed by J. Willard Gibbs in 1902 [7]. As noted earlier, Samuelson in 1947 [2], who drew the influence from Irving Fisher [54], drew on Gibbs’s approach for his reformulation of standard economic theory, a development much criticized by Mirowski [1], who derided all as economists exhibiting “physics envy”.

More recently, there have been a variety of economists using statistical mechanics to develop stochastic models of various economic dynamics, including work by Hans Föllmer in 1974 [55], and then in the 1990s, just as the econophysicists were getting going by Blume [56], Durlauf [31] (pp. 83–104) and [57], Brock [58], Foley [59], and Stutzer [60]. Stutzer applied the maximum entropy formulation of Gibbs with the conventional Black–Scholes model [43], drawing on Arrow–Debreu contingent claims theory [61]. Brock and Durlauf [62] would formalize the general approach within the context of socially interacting heterogeneous agents maximizing utility in a discrete choice setting.

To a substantial degree, most econophysicists were not aware of either the more recent work along these lines, much less the deeper work further in the past, with this leading to some of them making unfortunately exaggerated claims about the originality and transformative nature of what they were doing. These problems were discussed in a critical essay called “Worrying trends in econophysics” by Gallegati et al. in 2006 in *Physica A* [63]. They identified the following as problematic trends: missing knowledge of the existing economics literature, a readiness to believe there may be universal empirical regularities in economics not really there unlike in physics, much use of unrigorous statistical methods sometimes just looking at figures, and relying on inappropriate theoretical foundations such as invalid conservation principles. McCauley responded [64], taking a hard line, that economic theory is so worthless that it should be totally replaced by ideas coming from physics. Reviewing these arguments, Rosser [65,66] agreed that economists often make vacuous assumptions, despite excessively unreal assumptions damaging usefulness of models. One way to deal with this is to have more joint research between economists and physicists.

## 4. Forms of Entropy

In the Gibbsian statistical mechanics, the question of maximizing entropy is a crucial element, which leads us to the question of what entropy is. Its original formulation came from Ludwig Boltzmann [67], although it was not as many thought the form that appeared on his grave [68] that has long received a great deal of attention. The statistical mechanics problems involve aggregating out of individual molecular interactions to observe systemic averages, such as temperature out of such a motion in a space. Letting *S* be entropy, *k_B_* be the Boltzmann constant, and *W* be the statistical weight of the system macroscopic state (also known as the “thermodynamic probability”), then the following equation can be written as:*S* = *k_B_* ln *W*,(4)
where the configurational statistical weight of the macrostate of the system, *W*, defines the number of ways (configurations) of the arrangement of *N* of the identicalideal classical gas molecules in the microstates of the system (constituting a given macrostate), where *N_i_* is the number of the identical molecules in the microstate *i*. The author uses this physical interpretation later in the work, given *N* is the sum of over the *n* available microstates of the system each given by *N_i_*. Then according to Chakrabarti and Chakraborty [69], this implies that one is dealing with factorials multiplying each other as:*W* = *N*!/Π*N_i_*!. (5)

From this, Boltzmann entropy can be rewritten as:*S* = *k_B_* ln (*N*!/Π*N_i_*!). (6)

Moreover, the transition to the thermodynamics of an ideal classical gas at a temperature of T > 0 requires additional conditions to be taken into account, concerning the consistency of the total number of molecules of the gas, *N*, and the total energy, *E*, of all molecules.

Gibbs [7] famously declared that “mathematics is a language”, which indeed he viewed as applying to his analysis of entropy within statistical mechanics. However, while we can view the mathematical formulation of Boltzmann entropy as a linguistic matter, it describes the real physical phenomenon of thermodynamics. Thus, it can be viewed as being ontological entropy [70], as it can be applied to more abstract phenomena with less linkage to definite physical processes, thus allowing them to be labeled metaphorical entropy. The first application beyond thermodynamics was information patterns in the form of Shannon entropy [71]. This describes *H*—the probability distribution of informational uncertainty states for message i that reflects the whole set of information concerning the relevant microstate, *H*(*p*_1_…*p_n_*). Therefore, informational entropy involves adding up the individual log probabilities times their probabilities to give [71,72,73]:*H*(*p*_1_…*p_n_*) = −*k_B_*Σ*p_i_*ln*p_i_*. (7)

An obvious question arises as to how this widely used and influential metaphorical entropy measure relates to the ontological one of Boltzmann. In fact, they are proportional to each other as the number of possible states, *N*, approaches infinity, because *p_i_* = *N_i_*/*N*, resulting in [74,75]:*S* = *k_BN_*Σ*p_i_*ln*p_i_*. (8)

## 5. Ontological Entropy, Econophysics, and the Foundations of Growth

Ontological entropy lies at the heart of the econophysics foundation of economic growth due to the profound importance of energy both through the role of steam engines in industrial production and electricity and in agriculture through the thermodynamic transmission of solar energy through the larger global biosphere. The origin of understanding thermodynamics came from Sadi Carnot [76] in 1828 and later more fully Rudolf Clausius [77]. In 1971, Nicholas Georgescu-Roegen, [78] argued that the openness of the global biosphere to the sun allows temporarily overcoming the law of entropy [79]. Even so, there is a limit to solar energy, which implies limits for economic activity. However in an open system, anti-entropic forces can operate to develop order in local areas, drawing on the argument of Schrödinger [80] that life is ultimately an anti-entropic process involving the drawing of energy and matter from outside the living organism until it dies. Georgescu-Roegen also argued for this to extend to broader material resource inputs, subject to a form of the law of entropy. More broadly for Georgescu-Roegen [78] (p. 281), “the economic process consists of a continuous transformation of low entropy into high entropy, that is, into *irrevocable waste*, or, with a topical term, into pollution”.

Many ecological economists [81,82] have supported the idea of entropy as an ontological limit to growth. However, while this is clearly true, others have noted that the limit is many orders of magnitude above other limits that are more immediate [83,84,85]. Drawing down stored fossil fuel energy sources generates climate-changing pollution by releasing CO_2_ and thus further limiting growth. Others note the unlimited ingenuity of the human mind, with Julian Simon [86] (p. 347) arguing that “those who view the relevant universe as unbounded view the second law of thermodynamics as irrelevant to the discussion”.

## 6. Ontological Entropy and Economic Value

Another argument has seen ontological entropy as the fundamental source of economic value in a parallel to the labor theory of value. The earliest version of this dates to the turn of the twentieth century in arguments by “energeticist” physicists [87,88,89]. Julius Davidson [90] saw the economics law of diminishing returns based on the law of entropy, with the law of diminishing marginal returns, probably the only “economic law” that has no exception to it. Davis [91] claimed “economic entropy” underlies the utility of money, but Lisman [92] argued this is not how thermodynamics operates in physics. Samuelson [93] ridiculed such arguments as a “crackpot”, even as he drew on entropic ideas of Gibbs [7] and Lotka [81].

Lotka [81] (p. 355) himself noted limits to this argument: “The physical process is a typical case of ‘trigger action’ in which the ratio of energy set free to energy applied is subject to no restricting general law whatsoever (e.g., a touch of the finger upon a switch may set off tons of dynamite). In contrast with the case of thermodynamics conversion factors, the proportionality factor is here determined by the particular mechanism employed”. Georgescu-Roegen [78] saw value as ultimately coming from utility rather than entropy. Thus, most people value the high-entropy beaten egg more highly than the low-entropy raw egg, and nobody valuing low-entropy poisonous mushrooms, due to utility rather than entropy.

## 7. Thermodynamic Sustainability of Urban–Regional Systems

The ontological entropic analysis of urban and regional systems sees them driven by the second law of thermodynamics based on actual energy transfers as argued by Rees [94], Balocco et al. [95], Zhang et al. [96], Marchinetti et al. [97], and Purvis et al. [98]. Alan Wilson [99] reviews both ontological and metaphorical approaches to the entropic analysis of urban and regional systems.

Considering urban–regional systems as open and dissipative systems, experiences allows the analysis of sustainability, depending on their energy and material flows [81,100]. In open systems, entropy can rise or fall, as energy and materials flow into them, in contrast to closed systems where entropy can only rise. This is the key to Schrödinger’s [80] that life is an anti-entropic process with organisms drawing in energy-generating structure and order while life lasts. Anti-entropy is also known *exergy* [101] and also negentropy or “negative entropy”.

Three concepts to distinguish are *S_total_* as total entropy, *S_i_* as inside entropy, and *S_o_* as outside entropy. Assuming the statistical independence between both the internal states and the external states, then their dynamic relationship can be written as: d*S_total_*/d*t* = d*S_i_*/d*t* + d*S_o_*/d*t*, with d*S_i_*/d*t* > 0.(9)

Given that d*S_o_*/d*t* can be either sign, when negative with an absolute value greater than that of *S_i_*, then total entropy may fall as the system absorbs energy and materials creating order, with entropy increasing outside as waste and disorder leave the system. Wackernagel and Rees [102] state, “Cities are entropic black holes” implying, as they produce large ecological footprints, their sustainability becomes impaired.

The maximum amount of the useful work possible to reach a maximum entropy condition of zero has been called *exergy* by Rant [101] initially for chemical engineering. This term is essentially identical to the term “chemical potential” and also “Gibbs-free energy”. Rant’s original formulation holds, when *B* is the exergy, *U* is the internal energy, *P* is the pressure, *V* is the volume, *T* is the temperature, *S* is the entropy, *μ_i_* is the chemical potential of component *i*, and *N_i_* is the moles of component *i*, implying:*B* = *U* + *PV* − *TS* + Σ*μ_i_N_i_*.(10)

Recognizing that this is an isolated system implies:
(11)dB/dt ≤ 0 ↔ dS/dt ≥ 0.

The right-hand side of Equation (11) simply holds for an isolated system, from which we see the anti-entropic nature of exergy, determining the irreversible spontaneous time evolution (or “time arrow”).

Balocco et al. [95] consider exergy in construction and building depreciation in Castelnuovo Beardenga near Siena, Italy, relying on an adaptation by Moran and Sciubba [103] of Rant’s model. Studying particularly the input–output of the construction industry, it is seen that those built in 1946–1960 provide higher sustainability than newer ones.

Zhang et al. [96] use entropy concepts to study sustainable development in Ningbo, China, a city near Shanghai, relying on ideas in [95,102,104,105]. They examine both ontological and metaphoric information entropy measures, as they consider four distinct aspects. The first two are sustaining input entropy and imposed output energy, arising from production. The second two constitute the urban system’s metabolic functions, regenerative metabolism and destructive metabolism, which linked to pollution and its cleanup, a measure of environmental harmony. These contrast developmental degree and harmony degree, with the finding during the 1996–2003 period that these two went in opposite directions, with the developmental degree rising (associated with declining entropy) and the harmony degree falling (associated with rising entropy). Thus, we see Chinese urban development sustainability issues clearly.

The dependence versus autonomy of systems on their environment, derived from dissipative structures of open systems considered by Prigogine [100], was formulated by Morin [106] and then used by Marchinetti et al. [97]. This finds urban systems development between autarchy and globalization, either extreme unsustainable, advocating a balanced path they see urban–regional systems as ecosystems operating on energy flows [107] based on a complex wholes emerging out of interacting micro-level components [108].

## 8. An Anti-Entropic Econophysics Alternative in Urban–Regional Systems

Opposing this entropic version urban and regional systems structure is a power law version. In higher-level distributional systems, entropy ceases to operate and become irrelevant. This reflects a balance of entropic and exergetic forces operating in the relations and distributions within urban–regional systems [109].

Power-law distributions of econophysics reflect dominant anti-entropic forces [70], and urban size distributions seem to show these [22]. For the Pareto power-law distribution of city sizes [35], *P* is the population, *r* is the rank, with *A* and *α* are constants, implying:*rPr*^α^ = *P_1_*. (12)

For *α* = 1, the population of rank *r* is written as:*P_r_* = *P*_1_/*r*. (13)

This is the rank-size rule of Auerbach [110] from 1913 and generalized in 1941 as Zipf’s law, claimed to be applied to many distributions [47]. Since Auerbach [110] proposed it and Lotka [81] challenged it, there has been much debate regarding the matter. Many urban geographers [111] claim it is a universal law. Many economists have doubted this, saying there is no reason for it, even as urban sizes may show power-law distributions [112,113]. However, Gabaix [22] says Zipf’s law holds in the limit if Gibrat’s law is true with growth rates, independent of city sizes. 

US city size distributions seem to have shown power-law distributions from 1790 to the present, although not precisely following the rank-size rule (the size of Los Angeles is now larger than half the size of New York), according to Batten [112]. A meta-study of many empirical studies by Nitsch [114] finds widely varying estimates over these studies, although showing an aggregate mean of α = 1.08, near Zipf’s value. Berry and Okulicz-Kozaryn [111] say Zipf’s law strongly holds if one uses consistent measures for urban regions across studies, especially the largest ones for megalopolises. Anyway, city size distributions seem to be power-law-distributed, suggesting dominance by anti-entropic econophysics forces in this matter.

Long viewed as foundational for economic complexity, increasing returns may provide a basis for power-law distributional outcomes [115]. Three different kinds of these have been identified for urban systems: firm-level internal economies [116], external agglomeration between firms in a single industry providing localization economies [117], and external agglomeration economies across industries generating yet larger-scale urbanization economies [118].

Papageorgiou and Smith [119] and Weidlich and Haag [120] have shown that rising agglomeration economies can overcome congestion costs to manifest urban concentration. However, such models have been partially replaced by “new economic geography” ones emphasizing economies of scale appearing in monopolistic competition studied by Dixit and Stiglitz [121]. Fujita [122] first applied this approach to urban–regional systems, although Krugman [123] received much more attention for his version [124].

## 9. General Equilibrium Value and Metaphorical Entropy

Metaphorical Shannon entropy offers a different approach than Arrow–Debreu general equilibrium theory of value. Arrow and Debreu views equilibrium as a fixed point set of steady prices. However, in the reality of a stochastic world, equilibrium may be a probability distribution of prices that are constantly varying everywhere at any point in time for any commodity that can be modeled entropically. The Arrow–Debreu solution is a special case of Lebesgue measure in the space of outcomes. Initially conceived by Föllmer [55], Foley [59] developed it, followed by Foley and Smith [125].

Foley [59] assumes all possible transactions within an economy have equal probability, implying a statistical distribution of behaviors in the economy where a particular transaction is inversely proportional to the exponential of its equilibrium entropy price. This is a shadow price derived from a Boltzmann–Gibbs maximum entropy set. The special case when “temperature” is zero implies Walrasian general equilibrium. The solution is not necessarily Pareto optimal, and it allows for possible negative prices as Herodotus observed in ancient Babylonian bridal auctions, where they sold brides in descending prices that started out positive but then would go negative [126]. Foley emphasizes the crucial importance of constraints in this approach, as one finds in the Arrow–Debreu model.

If there are *m* commodities, *n* agents of type *k* who make a transaction *x* of which there is *h^k^[x]* proportion of agents type *k* out of *r*, which make transaction *x* out of an offer set *A*, of which there are *mn*, then *multiplicity W* of an assignment for *n* agents assigned to *S* actions, each of them *s*, which gives the probabilistic states across these possible transactions as:*W[n_s]_] = n*!*/(n_1_*!*…n_s_*!*...n_S_*!*)*. (14)

Shannon entropy of this multiplicity involves summing over these proportions similarly to Equation (7) and is written as:*H{h^k^[x]} = −Σ_k=1_^r^W^k^Σ_xeA_h^k^[x]*. (15)

This formulation maximizes entropy subject to certain non-empty feasibility constraints, thus giving the Gibbs solution:*H^k^[x] = exp[−Πx]/Σ_x_exp[−Πx],*(16)
with Π is the entropy shadow price vectors.

## 10. Metaphorical Entropic Financial Modeling

Schinkus [127] points out that econophysicists are more willing than most economists to approach data open to more possible distributions or parameter values, while favoring ideas from physics, including entropy for financial modeling. According to Dionisio et al. [128] (p. 161):

“Entropy is a measure of dispersion, uncertainty, disorder and diversification used in dynamic process, in statistics and information theory, and has been increasingly adopted in financial theory”.

Using the entropy law with Shannon or Boltzmann–Gibbs distributions can model distributions involving lognormality, both exhibiting normal Gaussian characteristics, Michael J. Stutzer [60,129] has drawn on both types of entropy to model Black–Scholes [43] formuli. In [129], he uses Shannon entropy, like Cozzolino and Zahner [130], allowing them to derive lognormal stock price distributions at the same time, similar to what Black and Scholes [43] did in deriving their options formuli without using entropy measures. Stutzer [129] considered a discrete form version modeling a stock market price dynamic by:**Δ*p/p = μ*Δ*t + σ√**Δ**t*Δ*z*, (17)
with *p* is the price, Δp is the random shock, Δ*t* is the time interval, and the second term on the right hand side is the random shock, distributed ~ *N*(0, *σ^2^*Δ*t*).

The order-maximizing solution for the neutral density of relative entropy-minimizing conditional risk given by the integral is written as:*arg min*_d*Q/*d*P*_*∫log* d*q/*d*p* d*q,*(18)
which satisfies a martingale restriction with *q* as a quantity:*r*Δ*t* − *E[(*Δ*p/p)(*d*q/*d*p)]* = *0*.(19)

Thus, the Black–Scholes option-pricing formula can be derived from a martingale product density arising from relative entropy minimizing conditional risk for an asset subject to IID normally distributed shocks. Stutzer understood this does not generate non-Gaussian distributions such as econophysics power law ones. He poses using Generalized Auto Regressive Conditional Heteroskedastic (GARCH) processes as an alternative.

More recent studies have expanded the forms of entropy used in studying financial market dynamics. Thus, transfer entropy has been used by Jizba et al. [131] to study differences in related financial times series focusing on spike events by Dimpli and Peter [132] to study cryptocurrency dynamics and by Kim et al. [133] for directional stock market forecasting. In addition, permutation entropy has been used in a variety of financial market econophysics applications [134].

## 11. Using Statistical Mechanics to Model Income and Wealth Distributions

Income and wealth dynamical systems can be driven by interactions between power-law distributions and non-power-law ones. Wealth dynamics apparently exhibit power-law distributions, while income distribution dynamics look to consist of entropy-related Boltzmann–Gibbs distributions. The former seem to drive the top 2–3 percent of income distributions, while the latter seem to drive income distributions below that level in the UK and US [28,40].

Entropy came to be used in generalizations of various income distribution measures as early as 1981, when Cowell and Kuga [135] presented a generalized axiomatic formulation for additive measures of income distribution. Adding two axioms to the standard model allowed a generalized entropy approach to subsume the well-known Atkinson [136] and Theil measures [137]. The former can distinguish the skewness of tails, while latter has more generality, with Bourgignon [137] showing the Theil to be the only zero-homogeneous decomposable “income-weighted” inequality measure. Adding a sensitivity axiom to others, Cowell and Kuga [135] argued a generalized entropy concept implies the Theil index, even as some argued that this linking was a challenge, with Montroll and Schlesinger [138] (p. 209) declaring

“The derivation of distributions with inverse power tails from maximum entropy formalism would be a consequence only of an unconventional auxiliary condition that involves the specification of the average of a complicated logarithmic function”.

It is unsurprising that both wealth and financial market distribution dynamics exhibit power-law distributions taking into account their close link, given Vilfredo Pareto’s [35] role in discovering them. Initially trained to be an engineer, Pareto came to study the dynamic social classes relations manifested by income distribution. He claimed a universally true pattern that held throughout “the circulation of elites” he studied, but he was wrong, with ironically his method superior for the study of wealth distributions. He claimed incorrectly that because of the constancy of the income distribution pattern, little can be performed to equalize income, because changes in political leadership simply substitutes one power elite by another with no income distribution change. However, large changes occurred, so his approach went “underground”, reappearing for other uses such as for urban metropolitan size distributions [111].

The sociologist, John Angle [139], revived using Pareto’s power-law distribution for studying income and wealth distribution dynamics starting in 1986. Then, econophysicists followed up with this, with their finding that wealth distributions follow Pareto’s power law view well [27,140,141].

The question arises as to whether we are dealing with ontological or “merely” metaphorical models in studying wealth and income distributional dynamics. Some see the stochastic elements in these distributions associated with thermodynamical processes fundamentally driving the distributional dynamics of income and wealth. However, these do not appear to be direct ontological processes as with Carnot’s steam engines. More likely, these reflect dynamics associated with no substantial changes in public distributional policies.

Yakovenko and Rosser [40] show a model with an entropic Boltzmann–Gibbs dynamics for lower-income distribution and a Paretian power-law distributions for higher-level income dynamics. There is an assumption of the conservation of money or income or wealth, which has not held in recent years as top-level incomes have exploded although it did much more so in earlier decades. This is consistent with lognormal entropic dynamics appropriate for the majority of the population below a certain level where wage dynamics predominate, while a Pareto power law is more appropriate for the top level whose income is more determined by wealth dynamics.

Assuming money conservation, *m*, the Boltzmann–Gibbs entropic equilibrium distribution has probability, *P*, with *m* seen as:*P*(*m*) = *ce*^−*m*/*Tm*^,(20)
with *c* is a normalizing constant, and *T_m_* is the “money temperature” thermodynamically, equaling the money supply per capita. The portion of the income distribution below about 97–98 percent seems to be well modeled by this formulation.

If there is a fixed rate of proportional money transfers equaling *γ,* then the Gamma distribution rather than the Boltzmann–Gibbs distribution better describes the stationary money distribution with a power-law prefactor, *m^β^*, such that:*β = −1 −* ln*2/*ln*(1 − γ)*. (21)

This Boltzmann–Gibbs version more simply relates to a power law equivalent than that posed by Montrell and Schlesinger [138]. The connection between the models of wealth and income distributions is described as:*P(m) = cm^β^e^−m/T^*.(22)

Letting *m* grow stochastically disconnects this outcome from the maximum entropy solution [142], so the stationary distribution becomes Fokker–Planck equation-driven mean field situation, not Boltzmann–Gibbs distribution, although inverse Gamma in [27,142] is a Lotka–Volterra form showing *w* as the wealth per person and *J* as the average transfer between agents, with σ being the standard deviation:*P(w) = c[(e^−J/σσw^)/(w^2+J/σσ^)]*. (23)

This model provides an empirical explanation of income distribution consistent with Marxist and other classical economic views of socio-economic class dynamics [41,42,143].

Figure 1 exhibits this in the log–log form for the 1997 US income distribution, with the Boltzmann–Gibbs section for the lower 97 percent of the distribution being nonlinear on the left-hand side, while the Pareto section is linear in logs on the right-hand side showing the top 3 percent of the income distribution (Figure 4.5 of [144]).

There has been further use of variations on the Gamma distribution in studying market dynamics, with Moghaddim et al. [145] using the Beta Prime distribution to study housing market inequality dynamics.

## 12. The Revenge of Metaphorical Entropy as Bubbles Crash

Financial market dynamics interact with income and wealth distribution dynamics during speculative bubbles following a Minsky process [146,147,148]. During major bubbles, the top portion of the income and wealth distributions rises noticeably relative to the lower portion. Anti-entropic dynamics drive this process and its reversal, when the bubble crashes, hence the “revenge of entropy”. Thus, during a bubble, this upward movement of the Paretian portion also moves its boundary with the Boltzmann–Gibbs portion leftward.

The Great Depression brought the end of the “Gilded Age” after a major financial crash that appears to have lowered the top end of the income distribution, as noted by Smeeding [149]. The 2007–2009 Great Recession had several different bubbles happening, leading to a more complex outcome, with the housing bubble crash badly hurting the middle class, while crashes of the stock market and derivatives markets predominantly hurt the wealthy. The US stock market fell from more than half its value to its bottom in 2009, with total wealth declining by 50 percent. Top 10 percent wealth declined by 13 percent, while top 1 percent wealth declined by 20 percent [149]. However, the stock market quickly turned around, rising more rapidly than in the 1930s or after 2000, while the US housing market grew more slowly. Thus, wealth inequality declined for a while during 2008–2009. It increased again after that as the rising stock market aided those at the top, while the continuing problems of the US housing market held back the middle class. This was the Minsky dynamic at work in a more complex form than seen at other times.

Support for this can be seen looking at the end of the dotcom bubble in 2000, even though somewhat weak, as indicated in Figure 2 (Figure 4.7) [144] showing the log–log relation for the US income distribution for the years 1983–2001, with further discussion in [150] and extension to a sample of 67 nations in [151]. Mostly, the Boltzmann–Gibbs section barely moved, but there were small annual changes in the Paretian part, manifesting gradually increasing inequality over time. However, there is an exception here, the change between 2000 and 2001, with 2000 being the end of the dotcom bubble. This time interval exhibited a reversal, with the 2001 Paretian portion lying below the 2000 portion. This is consistent with a revenge of entropy following the dotcom bubble crash, as the 1990s came to an end.

## 13. Conclusions

The term “econophysics” is of recent vintage, barely a quarter of a century old. However, the idea behind it that ideas and even laws of physics have strongly influenced economicss in a variety of ways is certainly correct. One of such ideas that has deep connections with the newer econophysics is the concept of entropy, which has been applied to many parts of economics, including general equilibrium theory, growth theory, business cycles, ecological economics, urban and regional economics, income and wealth distribution patterns, and financial market dynamics. Some of these applications are ontological in the sense of drawing directly on the second law of thermodynamics as the actual physical driving force involved, such as understanding energy flows through the biosphere and the economy from the Sun. Others are metaphorical, as they draw on models of information theory or other non-specifically physical models using the mathematics of entropy theory. Econophysics has also long emphasized the ubiquity of power-law distributions for many economic phenomena, which in some areas arise from anti-entropic processes that conflict with entropic tendencies. This can generate an underlying dynamic, with an especially dramatic example involving the dynamics of income distribution interacting with business cycles and related financial market dynamics.

## Figures and Tables

**Figure 1 entropy-23-01286-f001:**
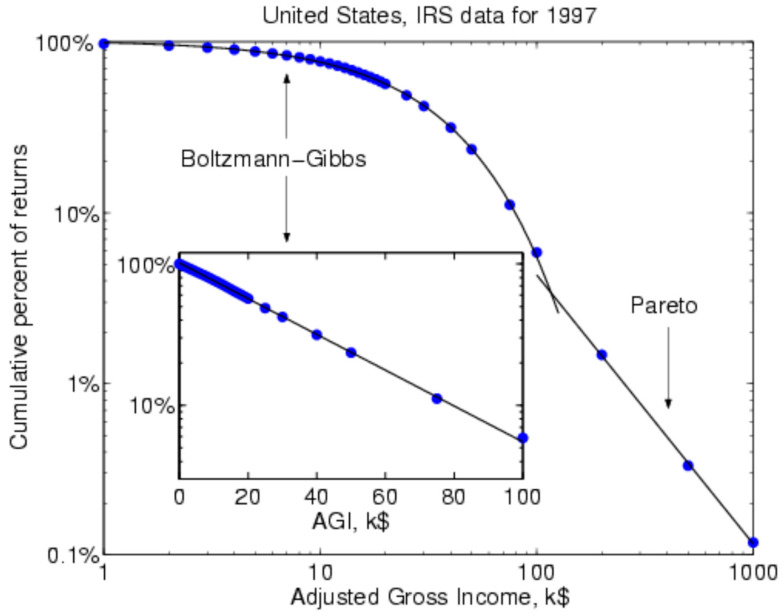
Log–log United States Income Distribution, Boltzmann–Gibbs, and Pareto Sections in 1997 from Yakovenko (Figure 4.6) [144].

**Figure 2 entropy-23-01286-f002:**
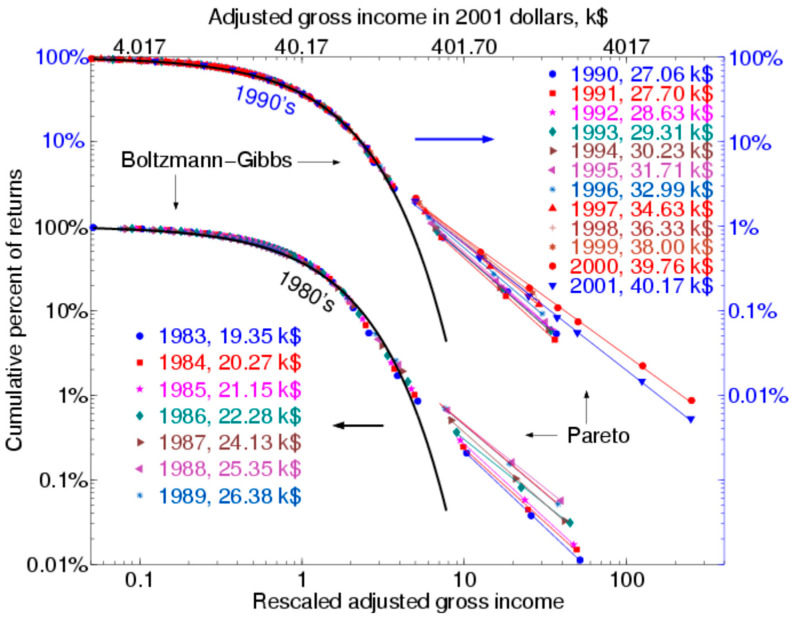
Log–log US Annual Income Distribution during 1983–2001 from Yakovenko (Figure 4.7) [144].

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
