# Peer review of "Econophysics and the Entropic Foundations of Economics"

_entropy, 2021, doi:10.3390/e23101286_

Round 1
Reviewer 1 Report
The paper presents an extensive review of applications of the concept of entropy in economics. Importantly, the paper distinguishes between two kinds of entropy: ontological and metaphorical. The former is the entropy used in physical thermodynamics, which governs energy transformations as the foundation for economic processes. The latter is the informational entropy, which, among other things, plays important role in financial dynamics and economic inequality. A particular strength of the paper is the depth and breadth of historical survey of the literature, from the 19th century to the present time, bridging economics, physics, and other disciplines. Thus I highly recommend the paper for publication.
Below is a list of technical typos, which can be easily correct, and a few additional references:
- Page 7, line 177. The reference number for Shannon entropy should be [72], not [73].
- Page 9, line 220. The reference to Samuelson should be [94], not [950.
- Page 10, line 243. "briong" should be "bring".
- Page 11, line 256. "N his formulation" should be "In his formulation".
- Page 11, line 268. The reference to Svirzhev should be [106], not [167]. The name is spelled as Svirzhev in the main text, but as Svirizhev in the reference.
- Page 294, line 294. The second reference to econophysics should be [111] by Rosser, not [110]. From this point on, all the references in the rest of the paper are shifted by 1: Auerbach [111] should be [112], urban geographers [112] should be [113], and so on to the end of the paper.
- Page 13, line 298. Equation (12) does not look correct as written. The correct Equation (13) does not quite follow from (12) for alpha=1.
- Page 16, line 357. Equation (15) does not look correct as written. If it is indeed Shannon entropy, it should not be equal to zero.
Here are some additional references for consideration of the author:
- V. M. Yakovenko, "Monetary economics from econophysics perspective",
Eur. Phys. J. Spec. Top. 225, 3313 (2016),
http://dx.doi.org/10.1140/epjst/e2016-60213-3
The figure in this paper may better illustrate author's argument about the revenge of metaphorical entropy. The figure can be obtained by downloading the source at https://arxiv.org/format/1608.04832.
- Yong Tao, Xiangjun Wu, Tao Zhou, Weibo Yan, Yanyuxiang Huang, Han Yu, Benedict Mondal, and Victor M. Yakovenko,
"Exponential structure of income inequality: evidence from 67 countries",
Journal of Economic Interaction and Coordination 14, 345 (2019)
https://doi.org/10.1007/s11403-017-0211-6
http://arxiv.org/abs/1612.01624
The paper illustrates the exponential (entropic) distribution of income for 67 countries around the world.
- Recently published book (March 5, 2021)
"Anthill Economics: Animal Ecosystems and the Human Economy"
by Nathanial Gronewold
https://www.amazon.com/Anthill-Economics-Animal-Ecosystems-Economy/dp/1633886522
The book is written from the perspective of biophysical economics and touches upon both ontological and metaphorical aspects of entropy, albeit not using this particular terminology.
Author Response
Thank you for your comments.
Reference numbers have all been fixed. Typos have all ben fixed. The proper spelling of Svirezhev is that, with him now only appearing by name in References as #106. Equations 12 and 13 have been fixed, and Equation 15 no longer is shown equaling zero. The two references have been cited and are now #s 151 and 152. However, I think the figures I have here are better than the ones in 151.
Reviewer 2 Report
This is basically a review article, very similar to Ref. 40 in the manuscript. It may be useful to readers as it contains mention of several topics and a large citation bank. I have several issues, however, that must be corrected before the paper is accepted for publication.
First, eq. (23) must be rewritten in the form of eq. (18) in Ref. 40, that is σσ must be replaced with σ2, and references to Bouchaud & Mezard and Solomion & Richmond must be added, as it was done in Ref. 40.
Second, contrary to what the author states, Fig. 1 in the manuscript is not the same as in Fig. 5 of Ref. 40. (Incidentally, the author cites instead Ref. 141, which is the book by Cockshott et al, and not just Yakovenko. However, the numbering convention for figures is different there - by chapters - so I have to assume that he meant Ref. 40). In Fig. 5 points appear as they might be fitted by a single distribution, whereas Fig. 1 exhibits a sharp break: that is why Pareto tail is tangential to the bulk of the distribution in Fig. 5, while it is decidedly not so in Fig. 1.
Lastly, it should be noted that the function given by eq. (23) is the inverse gamma function. It was first to put to use in econophysics about 20 years ago. Since then models have been generalized to Beta Prime, and even Generalized Beta Prime to describe wealth distribution and financial markets. BP contains gamma and inverse gamma functions as its limiting cases and GBP also includes generalized gamma and generalized inverse gamma. Lognormal can also be obtained as a special limit. I would recommend that the author widens his literature pool and includes more recent results.
Author Response
Thank you for your comments
Why equation must stay as it currently is is that making the change suggested would involve putting superscripts on superscripts. This is accurate. The papers suggested to be cited were already cited in the paper, but are cited here again as suggested in this context.
The figures are from Yakovenko #145 and have now been corrected to be labeled as 4.6 and 4.7 from that source. They are exactly as Yakovenko put them in this source.
It is now noted that equation 23 is an inverse gamma function and a cite of a paper from 2019 applying the generalized beta prime distribution, #146, is now added.
Reviewer 3 Report
The article deals with various aspects of thermodynamics, including entropy in its formulations by Boltzmann and by Shannon, in their relations with economics.
Although the article describes various aspects of those connections, it tries to encompass too much ground and thus does not go deep in any of the aspects. It also misses lots of progress that has been made using entropy or some measures derived from it (mutual information, transfer entropy, and others) that have been published in Entropy and on other journals.
It also has some mildly serious flaws in writing, like those described below. Most of them are connected with equations that are not sufficiently clear in their description or writing.
So, I do not recommend the manuscript to be published in Entropy.
1 – Line 167. Equation (5) is not fully explained. The product of factorials is not properly explained.
2 Line 180. Equation (7) is not fully explained.
- The author should use the Equation module of Word, since the displayed equations are in poor formatting.
- Line 355. There is some error in this displayed equation.
- Again, equations 15 and 16 are not explained in the text. All terms of every equation should be clearly described in the main text. The same is true to most of the equations that follow.
I’ve also spotted some other typos/mistakes that I cite below. My recommendations are to change, at the author’s discretion:
- 131 “including by by Hans” to “including the work by Hans”;
- 213 “as thru fundamental source of” to “as the fundamental source of”;
- 220 “Samuelson [950” to “Samuelson [95]”;
- 256 “N his formulation, if B is exergy” (better rewrite);
- 266 “and1946-1960” to “and 1946-1969 as”;
- 276 “second showsthe” to “second shows the”;
- 300 “Pr = P1/r,” to “Pr = P1/r.”;
- 366 “while faovring ideas from physics” to “while favoring ideas from physics”.
Author Response
Thank you for your comments.
I could find no application of mutual information entropy to econophysics or economics (there are ones to genetics). However, three citations have now been added for applications of transfer entropy, #s 132, 133, and 134, as well as a citation to applications of permutation entropy, which was published in Entropy in 2012, #135.
Explanations of the various equations requested have now been made. Equation 14 is accurate as is.
Also, all the typos have been fixed.
Round 2
Reviewer 1 Report
Satisfactory revisions have been made, so the paper can be now accepted.
Reviewer 2 Report
No suggestions
Reviewer 3 Report
I am sorry, but I stil consider the manuscript an overview of applications of entropy, but without any depth or novelty. So, I d not recommend it for publication.
This manuscript is a resubmission of an earlier submission. The following is a list of the peer review reports and author responses from that submission.